# Kalman-filter based stochastic-multiobjective network optimization and maximal-distance Latin hypercube sampling for uncertain inundation evacuation planning

Tsang-Jung Chang<sup>1</sup>, Yi-Hsuan Shih<sup>1</sup>, Ming-Che Hu<sup>1</sup>

<sup>5</sup> <sup>1</sup>Department of Bioenvironmental Systems Engineering, National Taiwan University, No. 1, Sec. 4, Roosevelt Road, Taipei 10617, Taiwan

Correspondence to: Ming-Che Hu (mchu@ntu.edu.tw)

**Abstract.** The subject of this research is to develop Kalman-filter based stochastic-multiobjective network optimization and maximal-distance Latin hypercube sampling methods regarding uncertain inundation evacuation planning. First, this research

- proposes a maximal-distance Latin hypercube sampling method to seek maximal space-filling sampling in uncertain flooding factor space. Uncertain inundation factors including upstream inflow, downstream water level, and channel friction resistance uncertainty are considered. Incorporated with the sampling method, HEC-RAS hydraulic model simulates stochastic flooding scenarios. Next, a Kalman-filter based stochastic-multiobjective network optimization model is established for uncertain inundation evacuation. Kalman-filter method iteratively predicts the flooding state of the next stage and updates prediction
- and decision according to new measurements. Kalman-filter based stochastic-multiobjective programming determines optimal shelter capacity expansion in the here-and-now stage and the best evacuation planning for each scenario in the wait-and-see stage. A case study of stochastic inundation evacuation in Muzha, Taiwan, is conducted. The contribution of this study is to incorporate Kalman-filter based stochastic-multiobjective network programming, HEC-RAS hydraulic simulation model, and maximal-distance Latin hypercube sampling to analyze inundation evacuation planning under uncertainty. The results show
- tradeoff between shelter expansion and evacuation time; furthermore, decreasing marginal effect of capacity expansion for evacuation time reduction is presented.

Keywords: Inundation evacuation, Kalman-filter, Stochastic-multiobjective network programming, Maximal-distance Latin hypercube sampling

# **1** Introduction

This study proposes an innovative framework for uncertain inundation evacuation planning. In this research, a Kalman-filter based stochastic-multiobjective network optimization model is newly developed for inundation evacuation under uncertainty. In addition, a maximal-distance Latin hypercube sampling method is established to simulate flooding uncertain scenarios. The maximal-distance Latin hypercube method samples upstream inflow, downstream water level, and channel friction resistance uncertainty by maximizing space-filling sampling of uncertain factor space. Incorporating with the maximal-distance sampling,

HEC-RAS hydraulic model is used to simulate stochastic flooding scenarios. Moreover, a case study of stochasticmultiobjective inundation planning in Muzha, Taiwan, is conducted. Possible flooding scenarios of Muzha are simulated. Then a tradeoff and uncertainty analysis of shelter capacity and inundation evacuation are presented.

Natural hazards, such as typhoons, hurricanes, and cyclones, lead to severe storms, heavy rainfall, and possible inundation events. Inundation might result in serious damage to people, property, and facilities (Parker and Fordham, 1996;

- Rodrigues et al., 2002; Romanowicz and Beven, 2003). Hence, inundation evacuation planning is an important consideration for preventing the loss of life and property damage (Li et al., 2012; Parker and Priest, 2012; Hegger et al., 2014; Zhang and Pan, 2014; Wang etal., 2015; Wood etal., 2016; Azam et al., 2017). To achieve inundation evacuation planning, locations and capacities of protection refuges should first be designed and constructed. Subsequently, decision support systems of emergency evacuation must be planned (Barbarosoglu and Arda, 2004; Bird et al., 2009; Taubenböck et al., 2009; Marrero et al., 2010;
- Yeo and Cornell, 2009; Bozorgi-Amiri et al., 2013; Pourrahmani et al., 2015; Xu et al., 2016; Hou et al., 2017; Liu et al., 2017; Muhammad et al., 2017). The majority of evacuation planning problems are formulated and solved as network programming models (Yi and Ozdamar, 2007; Stepanov and Smith, 2009; Bretschneider, 2012; Li et al., 2012; Fraser et al., 2014; Alaeddine et al., 2015). Such network programming models have been proposed by Ford and Fulkerson (1958), and these models are capable of determining the minimal distance or maximal flow transported from sources to sink nodes. Various types of network
- programming models are applied to evacuation planning. Maximal flow network models aim to evacuate and reallocate maximum evacuees within a given time. Minimal cost network problems seek to achieve minimal evacuation times to transport all evacuees to shelters. Shortest path network models present optimal evacuation routes for evacuees (Yi and Ozdamar, 2007; Bretschneider, 2012). Many related studies have discussed evacuation transportation system problems (Yi and Ozdamar, 2007; Stepanov and Smith, 2009; Abdelghany et al., 2014). Evacuation planning involves various uncertain factors, including
- unpredictable impacts, stochastic intensity, random locations of hazard, and uncertain responses of evacuees (Li et al., 2012; Yao et al., 2009). Because stochastic evacuation planning for uncertain flooding scenarios is an important consideration (Romanowicz and Beven, 2003; Barbarosoglu and Arda, 2004; Bozorgi-Amiri et al., 2013), stochastic programming models provide powerful tools for dealing with uncertain disaster and evacuation planning (Yao et al., 2009). Stochastic multi-stage programming models determine decisions at each stage. The most basic multi-stage programming model involves two stage
- programming models (Romanowicz and Beven, 2003; Barbarosoglu and Arda, 2004; Bozorgi-Amiri et al., 2013). Li et al. (2012) constructed bi-level programming models to determine the optimal capacity of shelters and evacuation route systems. Kongsomsaksakul et al. (2005) sought optimal locations and investments for shelters.

A Kalman-filter based stochastic-multiobjective optimization model is formulated to analyze the uncertain inundation evacuation problem. Kalman-filter is a control algorithm involving sequential prediction, measurement, update, and optimization. In each iterative step, once new measurement is conducted, Kalman-filter predicts the current state variables with uncertain noise. Then, the predictions are updated and the real-time optimization is analyzed. The optimization model is a stochastic two-stage programming model which can be extended to multi-stage programming if the decision making procedure involves more than two stages. The first stage decision of a two-stage evacuation problem includes the selection of

shelter locations, shelter capacities, transportation capacities, and other investment choices. The second stage decisions are concerned with evacuation planning. Furthermore, the proposed stochastic optimization considers multiple objective and the tradeoff among conflicting objectives is conducted. This study utilizes the weighting method to solve multiobjective optimization model to approximate the Pareto front of multiobjective problems. Then, the optimization model with a weighted objective function is readily solved. By applying weighting method, the general algebraic modeling system (GAMS) is used to solve the stochastic-multiobjective model.

- Previous studies have developed uncertain inundation simulations (Romanowicz and Beven, 2003; Mas et al., 2012). This study uses maximal-distance Latin hypercube method to sample uncertain factors and uses the HEC-RAS hydraulic model to simulate possible scenarios. HEC-RAS, developed by the Hydrologic Engineering Center (HEC), is a river analysis system (RAS). HEC-RAS solves continuity and momentum equations to simulate water flow condition of rivers and channels (Horritt and Bates, 2002; Knebl et al., 2005; Pappenberger et al., 2005). Latin hypercube is an efficient computational sampling method
- (McKay et al., 1979; van Dam et al., 2007; van Dam, 2008; Joseph and Hung, 2008; Husslage et al., 2010). This study proposes a modified Latin hypercube method with maximal total distance among sampling. The maximal-distance Latin hypercube method samples three uncertain factors including upstream inflow, downstream water level, and channel friction resistance. The sampling method is incorporated with HEC-RAS to simulate stochastic inundation scenarios. Then, the optimal evacuation planning under uncertainty is determined using the stochastic-multiobjective network programming model. To provide a
- spatial and temporal analysis for decision making, the results are displayed on the geographic information system (GIS) platform. Compared with related studies, the significant contribution of this research is to develop an innovative stochastic-multiobjective network optimization, maximal-distance Latin hypercube sampling, and HEC-RAS flooding simulation for uncertain inundation evacuation planning.

This paper is organized as follows. The Kalman-filter based stochastic-multiobjective inundation evacuation model and maximal-distance Latin hypercube sampling method are formulated and discussed in Section 2. A case study of Kalmanfilter based stochastic-multiobjective inundation evacuation planning in Muzha is analyzed in Section 3. In the section, the

# background of the uncertain inundation evacuation problem is introduced, and the results and discussion are presented. The conclusions are presented in Section 4.

# 2 Methods

This study newly establishes a Kalman-filter based stochastic-multiobjective programming model for analyzing inundation evacuation planning under uncertain scenarios. Maximal-distance Latin hypercube method is established to sample three uncertain factors, including upstream inflow (upper boundary condition), downstream water level (lower boundary condition), and friction resistance of channel. Uncertain flooding scenarios are simulated using the HEC-RAS. The uncertain flooding scenarios, the associate evacuation plans, and tradeoff analysis of multiple objectives are conducted and displayed on the GIS

(2)

platform. The framework of the Kalman-filter based stochastic-multiobjective network programming analysis of inundation evacuation planning is presented in Fig. 1.

A Kalman-filtered based stochastic-multiobjective network optimization model is constructed as follows. Nodes represent the source or sink points of the transportation network, such as shelters, inflow points, or outflow areas. Arcs represent transportation paths connecting two nodes, and each of these has its own transportation capacity. In this model, the

- arcs are bidirectional, indicating two way streets in the transportation network. In this paper, i and j indicate nodes of transportation networks, and k represents a node of shelter. The model also considers uncertain inundation scenarios, including possible damage to residential areas and transportation networks. The notation s and P(s) represent a stochastic scenario and its corresponding probability for uncertainty analysis. The evacuees are assumed to travel from the current node i to adjacent nodes nb(i) via connecting arcs. First-stage decisions for shelter and transportation capacity expansion before inundation
- uncertainty occurs are represented by  $y_1(k)$  and  $y_2(i, j)$ , respectively. Following the occurrence of uncertain flooding events, the optimal evacuation scheduling x(i, j, t, s) for the scenario s is determined in the second stage.  $C_s$  and  $C_t$  represent the expansion cost of shelter capacity and transportation capacity, and  $C_r$  is the evacuation cost. At time t, there are x(i, k, t, s)people to transport from node i to its neighbor shelter k, so the total evacuation time can be calculated as in Eq. (1). Eq. (1) determines the optimal evacuation plan by minimizing the expected total transportation time under uncertainty. The total
- investment cost is computed in Eq. (2). Furthermore, the weighting method multiplies each objective function by a weighting factor and sums up all weighted objective functions. Eqs. (1)-(2) is combined by the weighting method and the weighted multiobjective functions is presented in Eq. (3).

The constraint set is formulated as follows. CS(k) and CT(i, j, s) represent the existing capacities of shelter and transportation, respectively. The capacity constraints for shelters and transportation are established in Eqs. (4)-(5). Eq. (4)

- ensures that the total number of evacuees entering shelter k cannot exceed its existing capacity plus the expansion. Similarly, in Eq. (5) imposes that the number of people travelling on arc cannot be greater than its transportation capacity. Moreover, the mass balance constraints for each node are formulated in Eqs. (6)-(7). Here, r(i, t, s) is the number of evacuees to stay at node i, and IN(i, s) is the initial distribution of the evacuees at node i. At the initial time step (t = 1), the evacuees at node i can travel to other nodes. Eq. (6) stipulates that evacuees are allowed to either move to an adjacent node or remain at their current
- node. In the following time steps (t  $\neq$  1), people move on the network until they reach a shelters, according to Eq. (7). To ensure that all evacuees are evacuated, Eq. (8) demands that all evacuees must end up in a shelter. All decision variables should be positive, and non-negativity constraints are defined in Eqs. (9)-(10). The stochastic-multiobjective network system is formulated in Eqs. (1)-(10); the tradeoff analysis between the minimal evacuation time and the minimal total cost is conducted by weighting method with GAMS.

MIN
$$\sum_{s} \{ P(s) \times \sum_{t} \sum_{i \in nb(k)} \sum_{k}^{m} [t \cdot x(i, k, t, s)] \}$$
 (1)

MIN 
$$C_s \times \sum_{k=1}^{m} \{y_1(k)\} + C_t \times \sum_{i,j} \{y_2(i,j)\}$$

$$MIN \qquad C_{\rm r} \times \sum_{s} \{ P(s) \times \sum_{t} \sum_{i \in nb(k)} \sum_{k}^{m} [t \cdot x(i,k,t,s)] \} + C_{s} \times \sum_{k}^{m} \{ y_1(k) \} + C_{t} \times \sum_{i,j} \{ y_2(i,j) \}$$
(3)

Nat. Hazards Earth Syst. Sci. Discuss., https://doi.org/10.5194/nhess-2017-248 Manuscript under review for journal Nat. Hazards Earth Syst. Sci. Discussion started: 26 July 2017

| s.t. | $\sum_{i \in nb(k)} x(i, k, t, s) - y_1(k) \le CS(k)$                                                   | ∀ k, s                      | (4)  |
|------|---------------------------------------------------------------------------------------------------------|-----------------------------|------|
|      | $x(i, j, t, s) - y_2(i, j) \le CT(i, j, s)$                                                             | ∀ i, j, t, s                | (5)  |
|      | $\sum_{j \in nb(i)} x(i, j, t, s) + r(i, t, s) = IN(i, s)$                                              | $\forall i \notin k, t = 1$ | (6)  |
|      | $\sum_{j \in nb(i)} x(j, i, t - 1, s) + r(i, t - 1, s) = \sum_{k \in nb(i)} x(i, k, t, s) + r(i, t, s)$ | ∀i ∉ k, k, t ≠ 1            | (7)  |
|      | $\sum_{t}\sum_{k,i\in nb(k)} x(i,k,t,s) = \sum_{i}^{n} IN(i,s)$                                         | ∀k, s                       | (8)  |
|      | $y_1(k) \ge 0, y_2(i,j) \ge 0$                                                                          | ∀i, j, k, s                 | (9)  |
|      | $x(i, j, t, s) \ge 0, r(i, t, s) \ge 0$                                                                 | ∀i, j, k, s                 | (10) |

Sequential prediction, measurement, update, and optimization are conducted for each step of Kalman-filter based stochastic-multiobjective optimization model. Evacuation people IN(i, s) and undamaged evacuation routes CT(i, j, s) are uncertain input parameters of the model in Eqs. (1)-(10). Prediction of those uncertain parameters are iteratively updated by using new measurement in Eqs. (11)-(12). Accordingly, the evacuation routing x(i, j, t, s) and destination shelters r(i, t, s) are optimized in each step using Kalman-filter based stochastic-multiobjective network programming.

$$IN(i,s)|_{t}^{update} = f(IN(i,s)|_{t-1}^{prediction}, IN(i,s)|_{t-1}^{measure})$$
(11)

$$CT(i, j, s)|_{t}^{update} = f(CT(i, j, s)|_{t-1}^{prediction}, CT(i, j, s)|_{t-1}^{measure})$$
(12)

The Latin hypercube method is an efficient statistical sampling method, and this study proposes a maximal-distance Latin hypercube method for sampling and simulating uncertain inundation scenarios. Suppose that there are I1 uncertain factors, and that the intervals for each uncertain factor are divided into I2 sub-intervals. This method only generates one sample for each sub-interval of all the uncertain factors. Thus, I2 samplings are produced. Next, the method generates random permutations of [1,2,...,12] for each uncertain variable, and then the uncertain factor samplings are reordered by these permutations. The maximal-distance Latin hypercube method repeats the sampling procedures I4 times iteratively, and the total pairwise distances are calculated for each procedure. The coordinates of two uncertainty samplings i2 and i3 (i2, i3 = 1, 2, 3, ..., I2) are sam(i2, i1) and sam(i3, i1), so that the Euclidean distance between two samplings is

 $\sqrt{\sum_{i1}[sam(i2,i1) - sam(i3,i1)]^2}$ . Hence, the optimal sampling is decided by maximizing the summation of pairwise 150 Euclidean distance in Eq. (13).

MAX total distance = 
$$\sum_{i2,i3} \sqrt{\sum_{i1} [sam(i2,i1) - sam(i3,i1)]^2}$$
 (13)

The procedure with the longest distance is selected as the best sampling, because it spreads the sampling points in the best manner. The maximal-distance Latin hypercube method is applied to sample uncertain inundation factors. Then, uncertain 155 flooding scenarios are simulated by the HEC-RAS hydraulic model with uncertain factors. HEC-RAS is a hydraulic model for channel flow simulation and floodplain management analysis. HEC-RAS considers the continuity and momentum equations for networks of rivers and channels, Eqs. (14)-(15). Then, the direct step method is utilized to compute water surface profiles. Here, t and x represent time and flow direction. Furthermore, Q, A, and Y represent the water flow, cross section area, and

water depth of channels, respectively. In addition,  $q_{\ell 1}$  and  $q_{\ell 2}$  represent the lateral inflow and outflow, g is gravity,  $S_0$  is the 160 bed slope,  $S_f$  is the friction slope, and  $V_\ell$  is the average lateral flow velocity along the flow direction.

$$\frac{\partial A}{\partial t} + \frac{\partial Q}{\partial x} - q_{\ell 1} + q_{\ell 2} = 0$$

$$\frac{\partial Q}{\partial t} + \frac{\partial}{\partial x} \left(\frac{Q^2}{A}\right) - gA\left(S_0 - \frac{\partial Y}{\partial x} - S_f\right) - q_{\ell 1}V_{\ell} + q_{\ell 2}\left(\frac{Q}{A}\right) = 0$$
(14)
(15)

#### **3** Results and discussion

The case study applies the HEC-RAS hydraulic model to simulate uncertain inundation scenarios for Muzha in Taiwan; then, stochastic-multiobjective inundation planning is conducted (Fig. 2). The uncertainty and tradeoff of inundation evacuation efficiency and shelter expansion expenditure are analyzed. Muzha is surrounded by the Jingmei River on its east, south, and west sites. It is located in southern Taipei, Taiwan. The total area of Muzha is approximately 3.41 km<sup>2</sup>. The southern area of Muzha has highest population density. In this study, the HEC-RAS hydraulic model is applied to simulate uncertain inundation scenarios of the Jingmei River. The cross section geometry and characteristics of Jingmei River are obtained from Taiwanese

Water Resources Agency. The accuracy of the HEC-RAS model's representation of the real system has been validated. The validation shows that water inflow of 1554 cms, water level of 12.81 m, and manning's roughness coefficient of 0.025 yields accurate water level simulation of Jingmei River.

The transportation network system of Muzha is plotted in Fig. 2. The transportation system has 774 nodes and 1079 arcs. All of the arcs are bidirectional in the traffic network. Shelters are established to accommodate people, and save evacuees from danger. Most of the shelters are schools or regional activity centers, because these places often incorporate an auditorium and they are convenient locations for delivering to. According to the Disaster Prevention and Protection Organization of Taiwan, there are 17 shelters in the study site with various capacities. The capacities range from 10 to 300 people, and the total of 1277 residents in this area can be accommodated. Shelter locations and capacities are represented by grey circles in Fig. 2.

- Potential inundation overflow locations are simulated by comparing the water stage and levee height. Fig. 3 illustrates simulation of water stage and levee heights of the Jingmei River. In this study, areas within a 200-meter radius around potential overflow sites are regarded as evacuation zones. Inundation evacuation areas are considered under various stochastic scenarios and Fig. 4 depicts the potential overflow and inundation areas of Xinhai Road Sec 7, Hengkung Bridge, and Daonan Bridge in Muzha. The probability of each inundation scenario depends on the number of simulation for which the potential water stage exceeds the levee height. Based on the HEC-RAS simulation of 425 times at each location, the probabilities for three inundation
- areas (Xinhai Road Sec 7, Hengkung Bridge, and Daonan Bridge) are 0.43, 0.15, and 0.42, respectively. Six shelters are located within or close to the evacuation zones. The evacuation area of Xinhai Road Sec 7 area is 0.315 km<sup>2</sup> with a population of 8,000 people. Hengkung Bridge area is 0.069 km<sup>2</sup> with 2,000 people, and Daonan Bridge area is 0.089 km<sup>2</sup> with 2,100 people. This study assumed that people living on higher floors may not need to evacuate.

The stochastic-multiobjective network programming analysis is performed for inundation evacuation planning in 190 Muzha. In this study, the weighting method is applied for the multiobjetive analysis. The weighting method approximates the tradeoff between different objectives by gradually adjusting relative weightings. As a base case, assuming weighting of five and one for the shelter expansion cost and evacuation time, respectively (i.e.,  $C_s = 5$  and  $C_r = 1$ ), yields optimal solutions with a shelter expansion for 128 people in the Shihjian Activity Center and evacuation times of 55, 12, and 24 for the uncertain flooding scenarios. The results show that the Shihjian Activity Center is the only shelter that is required to be expanded. Xinhai 195 Road Sec 7 flooding scenario dominates the Shihjian Activity Center expansion, because the scenario involves the largest probability of occurrence, the largest inundation area, and the highest number of people to be evacuated. Since the flooding evacuation area of this scenario contains the highest number of residents, the western area of Muzha is the potential hot zone

for evacuation. For comparison, the shelter expansion and evacuation planning for the inundation scenarios in Xinhai Road Sec 7, Hengkung Bridge, and Daonan Bridge areas are plotted in Figs. 5-7.

The tradeoff analysis between the shelter expansion cost and the evacuation time is conducted as follows. The weightings of the multiobjective functions are adjusted to investigate the tradeoff between shelter expansion cost and evacuation time. Two cases with different weightings are analyzed for comparison in the following. Case 1 represents a lower weighting of the shelter expansion cost ( $C_s = 5$ ), and a higher weighting for the expansion cost ( $C_s = 10$ ) is simulated in Case 2. Since tradeoff between objectives depends only on relative weightings, both weightings of the evacuation time  $C_r$  are assumed to be one, i.e.,  $C_r = 1$ . The results for the shelters expansions in the two cases are compared in Figs. 8-9.

In the stochastic-multiobjective inundation planning, shelter expansions are determined in the first stage before inundations occur. For higher costs of shelter expansion (C<sub>s</sub>), it becomes more difficult to increase the shelter capacity. For Cases 1 and 2, there are four shelters that are expected to be expanded. In Case 1, the expansion locations are different for each scenario, especially for Scenario 1. Because of the larger inundation area of Scenario 1, more people must be evacuated. The largest expansion of a shelter is in the west of Muzha (Shihjian Activity Center). The number of extra people to be covered by expansions is 259. In Case 2, the capacity expansion of Shihjian Activity Center reduces to 116 people. In addition, a different location for shelter expansion replaces one in the east from Case 1. The location of the new shelter is in the midpoint of three scenarios. Thus, all evacuation scenarios can be benefit from this expansion.

Figs. 8-9 illustrate the evacuation planning on the transportation network system. In Case 1, it takes 55 minutes for 215 all evacuees to settle in a shelter for Xinhai Road Sec 7 area, and 11 minutes and 16 minutes for Hengkung Bridge and Daonan Bridge areas, respectively. In Case 2, the maximum evacuation times for Xinhai Road Sec 7 and Hengkung Bridge areas remain the same, but evacuation for Daonan Bridge area requires longer evacuation time of 23 minutes. This is obvious because Case 2 put higher weighting to minimize expansion cost, and evacuation time would receive less weighting. Hence, in Case 2, evacuees from Scenario 3 need to travel further to shelters.

The results indicate that a lower weighting of shelter expansion cost tends result in increases shelter capacities, rather than evacuating people to more distant shelters. Conversely, the case with a higher weighting for the expansion cost results in

less shelter expansion. Consequently, an increase in shelter expansion weighting (or cost) reduces the motivation for shelter expansion. Thus, evacuees will be required to travel further, rather than the close shelter being expanded. In addition, comparing flooding areas shows that Xinhai Road Sec 7 area dominates the shelter expansion, because it has highest population density and the most people to evacuate.

In this study, the weighting method is applied to approximate the tradeoff between different objectives by gradually adjusting relative weightings. Hence, the analysis fixes the weighting of transportation time to be one and the other weighting is gradually increased, i.e. (1:1), (2:1), (5:1), (6:1), (7:1), (8:1), and (10:1). To make it more clear, those seven cases determine the same optimal solutions with the weightings of (0.50:0.50), (0.67:0.33), (0.83:0.17), (0.86:0.14), (0.88:0.12), (0.89:0.11),

- and (0.91:0.09), if we assume the summation of weights is equal to one. The detailed information of tradeoff between shelter expansion and transportation time is illustrated in Fig. 10 and Table 1. A lower expansion weighting increases shelter expansion, and decreases evacuation time. Thus, evacuation distance and time are reduced by greater shelter expansion. On the contrary, a higher shelter expansion weighting deducts shelter expansion. Table 1 shows that when the objective ratio exceeds 7:1, the transportation time and shelter expansion will stay the same. By comparing Case 1 and Case 2, tradeoff analysis shows that
- shelter capacity expansion of 75 (=303-128) people decreases total evacuation time by 997 (=7255-6258) minutes. Case 1 and Case 0 demonstrates that capacity expansion of 501 (=804-303) people lowers evacuation time of 1513 (=6258-4745) minutes. Obviously, this presents the decreasing marginal effect of capacity expansion for evacuation time reduction.

# 4 Conclusions

During disaster events, people must be evacuated to prevent injury. Evacuation planning is analyzed for various disasters, such
as earthquake, hurricanes, inundations, and nuclear accidents. This study analyzes inundation evacuation under uncertainty. A
Kalman-filter based stochastic-multiobjective network model was newly established for iterative prediction, measurement,
update, and optimization of stochastic inundation simulation and evacuation. Maximal-distance Latin hypercube method has
been incorporated with HEC-RAS hydraulic modeling to increase sampling and computational efficiency of uncertain flooding
simulation. Accordingly, the tradeoff and uncertainty analysis of evacuation planning was conducted. The flooding scenarios,
shelter capacity expansion, and evacuation planning have been presented on the GIS platform for decision making.

The significant findings are listed in the following. The tradeoff analysis was used to investigate the impact of the expansion cost weighting. Tradeoff analysis shows that shelter capacity expansion of 75 people decreases evacuation time by approximately 1000 minutes and further capacity expansion of 500 people lowers evacuation time of 1500 minutes. Decreasing marginal effect of capacity expansion for evacuation time reduction is presented. We also notice that a reduction of the shelter

capacity expansion leads to shelter locations selection which are more convenient for all uncertain scenarios. In addition, the results demonstrate that an inundation scenario with a higher population density dominates decisions regarding shelter expansions and evacuation.

Uncertain inundation evacuation planning is important as the frequency and impacts of disaster events increases. Future studies should include economical, spatial, and temporal analysis for disaster simulation and evacuation planning. In 255 addition, optimal control and real-time stochastic evacuation under sequential disaster events, climate change, hydrological, and geological uncertainty should be further investigated.

#### Acknowledgments

The authors wish to thank the editors and anonymous referees for their thoughtful comments and suggestions. The authors are responsible for opinions and errors. This research was funded by the Ministry of Science and Technology, Taiwan, under Grant NSC-102-2313-B-002-054-MY3.

# Nomenclature

In this paper, indices and decision variables use lowercase letters. Uppercase letters indicate given coefficients. The indices, coefficients, decision variables, and their definitions are listed below.

| 265                                        | Indices                                       |                                                    |  |
|--------------------------------------------|-----------------------------------------------|----------------------------------------------------|--|
|                                            | i, j                                          | Nodes of transportation network, $i, j = 1, I$     |  |
|                                            | k                                             | Nodes of shelters, $k = 1, K$                      |  |
|                                            | nb(i)                                         | Adjacent nodes of node i                           |  |
|                                            | S                                             | Uncertain scenarios, $s = 1, S$                    |  |
| 270                                        | t                                             | Time, $t = 1, T$                                   |  |
|                                            | Coefficients                                  |                                                    |  |
|                                            | Cs                                            | Cost of shelter capacity expansion                 |  |
|                                            | C <sub>t</sub>                                | Cost of transportation capacity expansion          |  |
|                                            | C <sub>r</sub>                                | Cost of transportation time                        |  |
| 275                                        | CS(k)                                         | Shelter capacity of shelter k                      |  |
| CT(i, j, s) Transportation capacity from i |                                               | Transportation capacity from i to j for scenario s |  |
|                                            | IN(i,s)                                       | Inflow of node i for scenario s                    |  |
|                                            | Decision variables                            |                                                    |  |
|                                            | r(i, t, s)                                    | Stay in node i at time t in scenario s             |  |
| 280                                        | x(i, j, t, s)                                 | Transportation from i to j at time t in scenario s |  |
|                                            | y <sub>1</sub> (k)                            | Expand capacity of shelter k                       |  |
|                                            | Expand capacity of transportation from i to j |                                                    |  |

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
