# Peer review of "Kalman-filter based stochastic-multiobjective network optimization and maximal-distance Latin hypercube sampling for uncertain inundation evacuation planning"

_Natural Hazards and Earth System Sciences, 2017_

## Referee Comment (RC1) · J.M. Marrero (Referee) · 27 Sep 2017

Dr. José Manuel Marrero
REPENSAR
Camino de Orellana, N27-254 y Rafael Larrea
Quito, Ecuador

e-mail: josemallin@gmail.com

25 Sep. 2017

Dr. Kai Schröter
Natural Hazards and Earth System Sciences Editor

**R.E. Paper Ref: nhess-2017-248**

**Kalman-filter based stochastic-multiobjective network optimization and maximal-distance Latin hypercube sampling for uncertain inundation evacuation planning**

Dear Kai Schröter,

Many thanks for your invitation to review the above paper. I have found this paper of interest and value, particularly given the growing interest in the study of crisis and risk management. The author report an interesting analysis with some novel perspectives. However, I suggest a major revision and I provide some suggestions in order to improve the manuscript. Below I have outlined overarching comments, and then listed smaller and more specific edits. Should you or the authors have any further comments or questions, I would be happy to address these, and also review any further revisions.

**General comments**

In my opinion, the evacuation model is decontextualised of what a real evacuation process means. The authors do not explain or describe the evacuation scenarios they want to simulate and how the multiple factors comprised by them are addressed by the evacuation model. It is very difficult to develop an evacuation model flexible enough to be used and adapted to whatever situation. The evacuation scenarios and behaviours that can be addressed/simulated by this model should be described in the methodology in order to understand the model's limitations and scopes.

In this work, different methods have been presented, however, the interaction between them and their specific characteristic are not very well explained. I think the authors should start explaining the type of evacuations process they want to simulate, according to the natural hazard and the characteristic of the threatened area. Then they should follow describing the flood model more deeply and how it interacts with the evacuation model. And finally the evacuation model. I think it is important to separate this approaches first and then highlight the interaction between them in order to understand much better the presented work.

It would be also useful the use acronyms when referring to the developed models or methods throughout the text.

**Specific comments**

**Abstract**

**Line 8**. "*The subject of this research is to develop...*" *The subject of this research* is to present the development (I understand the work have been done).

**Introduction**

**Lines 25-32**. I suggest to move this paragraph below and start the introduction in the current line 33.

**Line 35.** This is just an opinion. I think the evacuation plan mainly prevents the loss of life, but I am not sure about property damage (specially in an urban context), unless the evacuee carry with all his properties, which is not common. If people must be evacuated is because they live in a hazard area and very sure their houses will be left behind.

**2. Methods**

Before explaining how the evacuation model work, authors should explain or described what kind of evacuation scenarios they want to simulate and for what kind of evacuation scenarios is this evacuation model useful.

I think some of the following questions should be addressed, first as a general issue and then in the example: Is the evacuation process a self evacuation where people make the decision about where they want to go? Is an assisted evacuation, where authorities evacuate people using public transport or other approaches? Are people following a pre-established evacuation plan, where shelters locations and the evacuation routes are perfectly known by people? Or is it a random process? How is the shelter chosen by the evacuee (how this decision making is managed or represented by the model)? There should be a theoretical connection between these important questions and how they are solved or addressed by the model.

Have the evacuation time intervals been included or considered in this model? To assess the total evacuation time the model should manage many factors and time intervals (as has been described in the bibliography, see for example Urbanik et al. (1980); Lindell (2008); Opper (2003) and Frieser (2004), Marrero et al, (2013)) Has this evacuation model such ability? If so, it should be very well explained. Other way, the results may be underestimate. For example, how do people know that a shelter is full? I think they only know that when they reach the shelter. Then, they need an extra time to find out where they can go (next shelter), which it will be conditioned by how the shelters spread the information about their capacity status. Has this consideration been included in the model in order to assess the evacuation time? Adapting an evacuation model to an specific context is also important to get the best results.

Here I think the authors should differentiate the use of the model during a real crisis or in the context of a long-term designing process of an emergency planing. In the former, the capacity of a shelter or even the transportation network can not be expanded easily during the evacuation, even if it is necessary (the transportation network can be modified with a very well evacuation planing designed in advance, but more time is needed to improve the shelters). Related with the transportation network capacity, how the evacuees travel from their home to the shelters? do they travel by car, bus or they go by walk?. This considerations affects the transportation network capacity and the travel time and they have not been addressed throughout the text.

Given the importance of the weighting method in this work, I think It should be better explained. What does the weighting method mean in the context of a real evacuation? What specific part of the evacuation process have been simulated by this method? If I understood, the efficiency of the evacuation strategy depends on the tradeoff analysis, that is, the relation between the shelter expansion cost and the evacuation time. For such relationship, people should know the emergency plan in advance (evacuation routes and shelters location and capacity), but what they will not know is whether the shelter capacity has been increased or not, and the next shelter they can go if the one they found first is totally full. Probably I do not understand very well why this factor is so critical if people do not know the information in advance? If authors want to predict the decision behaviour of evacuees that should be better explained.

In order to understand the proposed method, it would be helpful to add a Figure where all important elements are present (nodes, arcs. Etc.) with their attributes an letters (i, j, k. etc.), not as a flow chart but as a schematic drawing.

**Line 94**. "*...are conducted and displayed on the GIS...*" has the software been developed in a GIS? If so, which programming language, which GIS platform? Is it available on web?

**Line 101**. What do the authors mean by "*...The model also considers uncertain inundation scenarios*"? Flooding scenarios/areas with a very low probability of occurrence. If I understood, the evacuation model does not define the flooding area, that process is conducted by the flood models. If so, the evacuation model should evacuated the affected areas or whatever zone given as an input data, no matter how uncertain it is. Please, make more clear this question because it is not clear if the authors are referring to the flooding scenario probabilities or the evacuation scenario probabilities.

**Line 100**. Why are the arcs bidirectional?

**Line 103**. "*...decisions for shelter and transportation capacity...*" Who make the decision to expand shelters and roads capacity?, and why?. This is an strategy and it should be linked with a more global emergency plan.

**Line 105**. "*...Following the occurrence of uncertain flooding events*" Does it mean that the methodology is only useful for such uncertain flooding events and it cannot be applied for the most common flooding events.

**Line 109**. "*...optimal evacuation plan...*" I would say evacuation strategy, and apparently for one threatened area. Are the best routes chosen or just the available routes?

**Line 118**. "*...At the initial time...*" What does initial time mean here? Did the authors consider the response time, the warning time, etc.? When the evacuation order was given? Or the evacuation proses starts when everybody is ready. In such a case, the final result of the model should represent the travel time, not the evacuation time.

**Line 120**. "*...demands that all evacuees must end up in a shelter...*" Why must all evacuees end up in a shelter? In real evacuations it is very common that people evacuate with their relatives or in other places, outside the dangerous area, which means they could follow different routes. If all evacuees must reach a shelter, the authors should explain why (see comments above related with the description of the evacuation scenario).

**3. Results and discussion**

**Line 168.** "*...is applied to simulate uncertain inundation scenarios...*" Probably I do not understand well, but why are the results of the uncertain inundation scenarios compared to the normal behaviour of the river during a flood? Are they specific or common flooding scenarios? Were the flooding areas assessed using only flood models or they also include real scenarios based on past events?

**Line 173**. "*...The transportation network system of Muzha is plotted..*" Does the evacuation model address the total transportation network or just the main roads? That should be explained in the methodology and highlighted here.

**Line 175**. About shelters, are they known by people at risk? Have they been included in an evacuation plan? Do people know the shelter they should go or is it a free choice? Please, be more specific in the description of the evacuation model you are trying to simulate an how this kind of emergencies are managed in Muzha.

**Line 178.** "*...Shelter locations and capacities...*" Is the maximum capacity shown? Is the capacity

included in the management of the global emergency plan as a key factor?

**Fig. 4**. Please add an arrow to show the flow direction of the river.

**Line 184**. *"...Based on the HEC-RAS simulation of 425 times..."* Given the same water level and levee heights, how the HEC-RAS compute the variation in the results?

**Line 186**. Why are there shelters located in the evacuation area? Is there any problem with the local authorities in the design of the emergency strategies? Are this shelters considered by the evacuation model as sink nodes?

**Lines 188**. Why people living on higher floors do not need to be evacuated? Is it a question related with the capacity of the evacuation model or an evacuation plan design? Please, explain this important question, because people do not use to do what it is expected during an emergency situation (more people travelling could cause traffic jams) and an evacuation simulation never should be done for crisis management considering just part of the people living in a threatened area.

**Line 193**. *"...evacuation times of 55, 12, and 24* **minutes**?

**Line 194.** *"... Shihjian Activity Center is the only shelter that is required to be expanded..."* that assuming that no one living above the first floor will make the decision to evacuate.

**Figs. 5-7**. Some questions here: first, how do you represent such expansion on figures?, because I cannot see them easily (even comparing with figure 8-9). Is the minimum capacity of the shelters what it is showed on figures or is the maximum capacity they can reach? Authors do not highlight the total capacity of the shelter, I mean, the capacity in normal condition and the maximum capacity in extreme condition that shelters could reach. Second, it is seems the authors have separated the flooding scenarios as if they were isolated flooding areas instead of a global process. How is that possible? If there is a flooding, I suppose that all flooding areas should be affected more or less at the same time, if not, authors should explain why. I say that because evacuees are using roads that they should not be (for example, the road located in the south that cross the Hengkung Bridge flooding area to reach the shelter located at west, fig 7). Is that possible because the evacuation start before the flooding? Third, you should keep highlighted the flooding areas in order to see from where the evacuee are coming from in each scenario, when a security area is reached and if there are more hazard areas that should be avoided by the evacuees. Fourth, has the evacuation model the capacity to assess the time needed to evacuate the flooding hazard zone or just the time to complete the travel time.

**Line 219**. *"...further to shelters..."* here, do the authors mean that people have to travel far away to find shelters because the closer ones were not expanded and they are full?.

**4. Conclusions**

**Line 239**. *"...Evacuation planning is analyzed for various disasters..."* Analyzed or used?

**Line 240.** *"... inundation evacuation under uncertainty..."* Again, does it means the framework developed is not valid for more common scenarios whit less uncertainty.

**Line 245**. *"...and evacuation planning have been presented on the GIS platform..."*. If the presentation on the GIS is oriented to decision makers, all figures should keep the hazard areas (authors should not delete the hazard limits just to show the results of the evacuation model). Decision makers need to understand the global process and where are located the threatened areas (all of them).

Should you require any further information, please do not hesitate to contact me.

Yours Sincerely,

José M. Marrero

---

## Referee Comment (RC2) · Anonymous Referee #2 · 10 Oct 2017

GENERAL COMMENT

This manuscript introduces new concepts to develop the evacuation scenarios considering the uncertainty of inundation evacuation planning. For inundation evacuation under uncertainty, a Kalman-filter based stochastic-multiobjective network optimization model is implemented. Moreover, a maximal-distance Latin hypercube sampling method is used to simulate uncertain scenarios of flooding. A case study in Muzha, Taiwan is then used to apply the new frameworks of evacuation scenarios. The results of implementing these new methods are then discussed.
[Figure]

Although the manuscripts have presented new frameworks to improve the quality of evacuation scenarios under uncertainty, in general, several points need to be improved: (1) The flow of writing is not well presented. Some parts need to be connected with a proper story. The writing seems to be inconsistent considering the flow of the story; (2) The main concept of evacuation used in this study has not been addressed in the text; (3) the contribution/novelty of this concept compared to the existing methodologies has not been discussed clearly. Subsequently, further revisions need to be carried out.

SPECIFIC COMMENT

ABSTRACT: line 12: channel friction resistance uncertainty ==> "uncertainty" can be deleted since you already used UNCERTAIN INUNDATION FACTORS

Line 15: THE new measurement?

The story flow of sentences in the last sentence of abstract (line 19-21) is not well connected.

INTRODUCTION

In my opinion, the problem in the second paragraph (starting from line 33) must be moved in the first paragraph. Therefore, the story will be something like this: general problem, specific evacuation problem, issues in the existing evacuation methodologies developed in the current literature, then addressed why your frameworks are necessary.

Line 48: MANY RELATED STUDIES HAVE DISCUSSED EVACUATION TRANS-PORTATION SYSTEM PROBLEMs ==> What is the problem? need to discussed clearly so the contributions of this study can be well presented.

LINE55: WHAT are the two stage of programming models? you then explained in Line 63-64 but the sentences in line55 seem to be unfinished.

METHODOLOGY

First, what type of evacuation that you present in this manuscript? is this self-evacuation process where the people are decided by themselves where they want to evacuate? or is there any pre-existing road/plan that has been developed before? Is there any announcement from the authorities for the evacuation?

Second, in principle, the evacuation time is calculated by summing initial reaction time (IRT) and evacuation time (ET). Three components are further considered to calculate the initial reaction time (IRT) including institutional decision time (DT), institutional notification time (NT), and reaction time of the community (RT). Do you consider this concept or everything has been included in the multiobjective optimization introduced in this study?. If it has been included, please state it clearly.

RESULTS AND DISCUSSION

In my opinion, the manuscript needs to discuss the comparison between the results from your study and the other existing frameworks that have been validated and used for another study/region. In this section, the results from this study have not been validated with the other methodologies/data.

Line 180: why 200-m is the radius of potential overflow? please clarify

Line 184: why the simulation is only 425 times?

Lines 185: Why only three probability scenarios presented in this section? are there only three cases? please clarify. Why are these three probs chosen?

Line 188: How to define the people live on higher floors? do you have all the building data in this areas? please explain it clearly.

Line 198: HOT ZONE changes to CRITICAL zone?

Line 208-209: the sentences are not clear. Suggest to re-write.

The comments for Figs 8-9 need to elaborate on the concept of calculating the evacuation time.

CONCLUSIONS

First and second sentences (line 239-240) is not necessery. Suggest to change or delete it.

The limitations of this study also need to be explained clearly in this section. Only future studies are presented but it can be connected with the limitations of this study.

---

## Author Comment (AC1) · 13 Dec 2017

Journal: NHESS

Title: Kalman-filter based stochastic-multiobjective network optimization and maximal-distance Latin hypercube sampling for uncertain inundation evacuation planning

MS No.: nhess-2017-248

MS Type: Research article

**Referee #1**

J.M. Marrero (Referee)

josemarllin@gmail.com

Dear Kai Schroter,

Many thanks for your invitation to review the above paper. I have found this paper of interest and value, particularly given the growing interest in the study of crisis and risk management. The author report an interesting analysis with some novel perspectives. However, I suggest a major revision and I provide some suggestions in order to improve the manuscript. Below I have outlined overarching comments, and then listed smaller and more specific edits. Should you or the authors have any further comments or questions, I would be happy to address these, and also review any further revisions.

**[Authors]:**

**Dear Referee,**

**Thank you for the comments on our manuscript. We have modified the manuscript accordingly. A detailed point-by-point reply to all of the comments are provided in the following.**

**Yours,**

**Ming-Che Hu, Yi-Hsuan Shih, Tsang-Jung Chang**

General comments

In my opinion, the evacuation model is decontextualised of what a real evacuation process means. The authors do not explain or describe the evacuation scenarios they want to simulate and how the multiple factors comprised by them are addressed by the evacuation model. It is very difficult to develop an evacuation model flexible enough to be used and adapted to whatever situation. The evacuation scenarios and behaviours that can be addressed/simulated by this model should be described in the methodology in order to understand the model's limitations and scopes.

In this work, different methods have been presented, however, the interaction between them and their specific characteristic are not very well explained. I think the authors should start explaining the type of evacuations process they want to simulate, according to the natural hazard and the characteristic of the threatened

area. Then they should follow describing the flood model more deeply and how it interacts with the evacuation model. And finally the evacuation model. I think it is important to separate this approaches first and then highlight the interaction between them in order to understand much better the presented work.

It would be also useful the use acronyms when referring to the developed models or methods throughout the text.

**[Authors]:**

**Thank you for the comments. Evacuations process of this research is further explained as follows. The assumption and limitation of the flooding simulation and evacuation models are discussed in the methodology section; then interaction between models are addressed. Simulation of the flooding scenarios is described in the case study section.**

*This study establishes a Kalman-filter based stochastic-multiobjective network optimization (KASMNO) model for analyzing both long-term and short-term inundation evacuation strategies. This KASMNO model determines (1) long-term shelter and transportation capacity expansion plans for authorities and (2) short-term evacuation routing for evacuees under flooding scenarios. For short-term evacuation procedures, authority decision, announcement, community reaction, and evacuation are considered. To simulate flooding scenarios, maximal-distance Latin hypercube method is used to sample three uncertain factors, including upstream inflow (upper boundary condition), downstream water level (lower boundary condition), and friction resistance of channel. Then potential flooding scenarios are simulated using the HEC-RAS hydraulic model and uncertain factors. The stochastic evacuation model can be further simplified to be a deterministic model by inputting deterministic inundation scenario.*

*Notice that the proposed model assumes that evacuees have the complete information about shelter capacity status and people would follow authority's evacuation plan. Otherwise, additional time needs to be estimated while people's individual behaviour exists under incomplete information cases. The weighting method is used to analyze the tradeoff between the shelter expansion cost and the evacuation time. The uncertain flooding scenarios, the associate evacuation plans, and tradeoff analysis of multiple objectives are conducted and displayed on the GIS platform. The framework of the Kalman-filter based stochastic-multiobjective network programming analysis of inundation evacuation planning is presented in Fig. 1.*

*The HEC-RAS hydraulic model is used to simulate uncertain water stage of the Jingmei River in Muzha. Uncertain upstream flow (0%, ±7%, ±14%), downstream water level (0%, ±6%, ±12%), and channel roughness coefficient (ranging from 0.013 to 0.045 by interval of 0.002) are considered in the model. Fig. 3 plots uncertain simulation of water stage of the Jingmei River. In this study, potential inundation overflow locations are determined by comparing the water stage and levee height. Then areas within a 200-meter radius around potential overflow sites are regarded as evacuation zones. Accordingly, Fig. 4 displays the three cases of overflow location and inundation evacuation areas including three cases of Xinhai Road Sec 7, Hengkung Bridge, and Daonan Bridge in Muzha. The probability of each inundation scenario depends on the number of simulation for which the potential water stage exceeds the levee height. Based on the HEC-RAS simulation at each location, the probabilities for three inundation areas (Xinhai Road Sec 7, Hengkung Bridge, and Daonan Bridge) are 0.43, 0.15, and 0.42, respectively. People can be evacuated to six shelters located close to the inundation zones. The evacuation area of Xinhai Road Sec 7 area is 0.315 km². Hengkung Bridge area is 0.069 km², and Daonan Bridge area is 0.089 km². Data of people living on each floor are not available. This case study assumes that people living on the first floor needs to be evacuated; the rest of people living on upper floors are not evacuated due to shelter capacity. The proposed stochastic-multiobjective model can further develop complete evacuation plans for all of people in threatened area by assuming whole area as evacuation places.*

Specific comments
Abstract
Line 8. "The subject of this research is to develop..." The subject of this research is to present the development (I understand the work have been done).
**[Authors]:**
**Thank you. The sentence is modified as follows.**

*The subject of this research is to present the development of Kalman-filter based stochastic-multiobjective network optimization and maximal-distance Latin hypercube sampling methods regarding uncertain inundation evacuation planning.*

Introduction
Lines 25-32. I suggest to move this paragraph below and start the introduction in the

current line 33.

**[Authors]:**

**Thanks for the comments. The Introduction section is reorganized. The first paragraph starts with introducing natural hazards and inundation and the purpose of this research is addressed in the following. Details change are modified in the updated manuscript.**

> *Natural hazards, such as typhoons, hurricanes, and cyclones, lead to heavy rainfall, severe storms, and then possible inundation events. Inundation might result in serious damage to people, property, and facilities (Parker and Fordham, 1996; Rodrigues et al., 2002; Romanowicz and Beven, 2003). Hence, inundation evacuation planning is an important consideration for preventing the loss of life (Li et al., 2012; Parker and Priest, 2012; Hegger et al., 2014; Zhang and Pan, 2014; Wang etal., 2015; Wood etal., 2016; Azam et al., 2017). To achieve inundation evacuation planning, locations and capacities of protection refuges should first be designed and constructed. Subsequently, decision support systems of emergency evacuation must be planned (Barbarosoglu and Arda, 2004; Bird et al., 2009; Taubenböck et al., 2009; Marrero et al., 2010; Yeo and Cornell, 2009; Bozorgi-Amiri et al., 2013; Pourrahmani et al., 2015; Xu et al., 2016; Hou et al., 2017; Liu et al., 2017; Muhammad et al., 2017).*

> *This study proposes an innovative framework for uncertain inundation evacuation planning. In this research, a Kalman-filter based stochastic-multiobjective network optimization (KASMNO) model is newly developed for inundation evacuation under uncertainty. In addition, a maximal-distance Latin hypercube sampling method is established to simulate flooding uncertain scenarios.*

Line 35. This is just an opinion. I think the evacuation plan mainly prevents the loss of life, but I am not sure about property damage (specially in an urban context), unless the evacuee carry with all his properties, which is not common. If people must be evacuated is because they live in a hazard area and very sure their houses will be left behind.

**[Authors]:**

**We agree with the comment. The sentences are modified as follows.**

> *Natural hazards, such as typhoons, hurricanes, and cyclones, lead to heavy rainfall, severe storms, and then possible inundation events. Inundation might result in serious damage to people, property, and facilities (Parker and Fordham, 1996; Rodrigues et al., 2002; Romanowicz*

*and Beven, 2003). Hence, inundation evacuation planning is an important consideration for preventing the loss of life (Li et al., 2012; Parker and Priest, 2012; Hegger et al., 2014; Zhang and Pan, 2014; Wang etal., 2015; Wood etal., 2016; Azam et al., 2017).*

2. Methods

Before explaining how the evacuation model work, authors should explain or described what kind of evacuation scenarios they want to simulate and for what kind of evacuation scenarios is this evacuation model useful.

**[Authors]:**

**Thank for the comment. The evacuation model and scenarios are addressed in the updated manuscript as follows.**

*This KASMNO model determines (1) long-term shelter and transportation capacity expansion plans for authorities and (2) short-term evacuation routing for evacuees under flooding scenarios. For short-term evacuation procedures, authority decision, announcement, community reaction, and evacuation are considered. To simulate flooding scenarios, maximal-distance Latin hypercube method is used to sample three uncertain factors, including upstream inflow (upper boundary condition), downstream water level (lower boundary condition), and friction resistance of channel. Then potential flooding scenarios are simulated using the HEC-RAS hydraulic model and uncertain factors.*

*The KASMNO model is constructed as follows. The model has two optimization stages including evacuation capacity expansion and evacuation routing. In the first stage, authority determines optimal expected solutions for shelter and transportation capacity expansion under flooding uncertainty. In the second stage, optimal evacuation routing is solved for each potential inundation scenario.*

I think some of the following questions should be addressed, first as a general issue and then in the example: Is the evacuation process a self evacuation where people make the decision about where they want to go? Is an assisted evacuation, where authorities evacuate people using public transport or other approaches? Are people following a pre-established evacuation plan, where shelters locations and the evacuation routes are perfectly known by people? Or is it a random process? How is the shelter chosen by the evacuee (how this decision making is managed or represented by the model)? There should be a theoretical connection between these important questions and how they are solved or addressed by the model.

**[Authors]:**

The model assumes that evacuation procedures involves authority decision, announcement, community reaction, and evacuation. The model assume that community follows the authority evacuation plan. The case study assumes that evacuees travel from their home to the shelters by walk. However, other transportation approaches can be considered and simulated by the model.

*This study establishes a KASMNO model for analyzing both long-term and short-term inundation evacuation strategies. This KASMNO model determines (1) long-term shelter and transportation capacity expansion plans for authorities and (2) short-term evacuation routing for evacuees under flooding scenarios. For short-term evacuation procedures, authority decision, announcement, community reaction, and evacuation are considered.*

*Notice that the proposed model assumes that evacuees have the complete information about shelter capacity status and people would follow authority's evacuation plan. Otherwise, additional time needs to be estimated while people's individual behavior exists under incomplete information cases.*

Have the evacuation time intervals been included or considered in this model? To assess the total evacuation time the model should manage many factors and time intervals (as has been described in the bibliography, see for example Urbanik et al. (1980); Lindell (2008); Opper (2003) and Frieser (2004), Marrero et al, (2013)) Has this evacuation model such ability? If so, it should be very well explained. Other way, the results may be underestimate. For example, how do people know that a shelter is full? I think they only know that when they reach the shelter. Then, they need an extra time to find out where they can go (next shelter), which it will be conditioned by how the shelters spread the information about their capacity status. Has this consideration been included in the model in order to assess the evacuation time? Adapting an evacuation model to an specific context is also important to get the best results.

[Authors]:

This research considers authority decision, announcement, community reaction, and evacuation. The model is able to evacuate community for deterministic and uncertain flooding events. This research assumes that the community follows the proposed evacuation plan while evacuation decision is made by the authority. Extra evacuation travel time needs to be added if people don't the complete information about shelter capacity status. The section is modified as follows. In addition, related studies are reviewed and cited.

*This KASMNO model determines (1) long-term shelter and transportation capacity expansion plans for authorities and (2) short-term evacuation routing for evacuees under flooding scenarios. For short-term evacuation procedures, authority decision, announcement, community reaction, and evacuation are considered (Urbanik et al., 1980; Opper, 2003; Frieser, 2004; Lindell, 2008; Marrero et al, 2013).*

*The proposed model assumes that evacuees have the complete information about shelter capacity status and people would follow authority's evacuation plan. Otherwise, additional time needs to be estimated while people's individual behaviour exists under incomplete information cases.*

- *Frieser, B. I.: Probabilistic evacuation decision model for river floods in the Netherlands, TU Delft master thesis, 2004.*

- *Lindell, M. K.: EMBLEM2: An empirically based large scale evacuation time estimate model, Transportation Research Part A: Policy and Practice, 42 (1), 140-154, 2008.*

- *Marrero, J. M., García, A., Llinares, A., la Cruz-Reyna, S. D., Ramos, S., and Ortiz, R.: Virtual tools for volcanic crisis management, and evacuation decision support: applications to El Chichón volcano (Chiapas, México), Nat. Hazards, 68: 955-980. https://doi.org/10.1007/s11069-013-0672-4, 2013.*

- *Opper, S.: Emergency planning for the Hawkebury Nepean Valley, 40th Annual conference of the floodplain management, 2003.*

- *Urbanik, T., Desrosiers, A., Lindell, M. K., and Schuller, C. R.: Analysis of techniques for estimating evacuation times for emergency planning zones, Battelle Human Affairs Research Centers Report No. BHARC-401/80-017, U.S. Nuclear Regulatory Commission, Washington, D.C., 1980.*

Here I think the authors should differentiate the use of the model during a real crisis or in the context of a long-term designing process of an emergency planing. In the former, the capacity of a shelter or even the transportation network can not be expanded easily during the evacuation, even if it is necessary (the transportation network can be modified with a very well evacuation planning designed in advance, but more time is needed to improve the shelters). Related with the transportation network capacity, how the evacuees travel from their home to the shelters? do they travel by car, bus or they go by walk?. This considerations affects the transportation network capacity and the travel time and they have not been addressed throughout

the text.

**[Authors]:**

**The problem we want to solve, the model, the scenarios, and the limitation are further addressed in the updated manuscript. The paragraphs are modified and the questions are addressed as follows.**

> *This KASMNO model determines (1) long-term shelter and transportation capacity expansion plans for authorities and (2) short-term evacuation routing for evacuees under flooding scenarios. For short-term evacuation procedures, authority decision, announcement, community reaction, and evacuation are considered.*

> *The case study assumes that evacuees travel from their home to the shelters by walk. However, other transportation approaches can be considered and simulated by the model.*

> *The model has two optimization stages including evacuation capacity expansion and evacuation routing. In the first stage, authority determines optimal expected solutions for shelter and transportation capacity expansion under flooding uncertainty. In the second stage, optimal evacuation routing is solved for each potential inundation scenario.*

Given the importance of the weighting method in this work, I think It should be better explained. What does the weighting method mean in the context of a real evacuation? What specific part of the evacuation process have been simulated by this method? If I understood, the efficiency of the evacuation strategy depends on the tradeoff analysis, that is, the relation between the shelter expansion cost and the evacuation time. For such relationship, people should know the emergency plan in advance (evacuation routes and shelters location and capacity), but what they will not know is whether the shelter capacity has been increased or not, and the next shelter they can go if the one they found first is totally full. Probably I do not understand very well why this factor is so critical if people do not know the information in advance? If authors want to predict the decision behaviour of evacuees that should be better explained.

**[Authors]:**

**Yes, the weighting method is used by authorities to analyze the tradeoff between the shelter expansion cost and the evacuation time. The proposed stochastic-multiobjective model assumes that evacuees have the complete information about shelter capacity status and people would follow authority's evacuation plan. Notice that extra evacuation travel time needs to be estimated if people don't have the complete information about shelter capacity status. The paragraph is modified as**

**follows.**

> *The weighting method is used to analyze the tradeoff between the shelter expansion cost and the evacuation time. The uncertain flooding scenarios, the associate evacuation plans, and tradeoff analysis of multiple objectives are conducted and displayed on the GIS platform. The framework of the Kalman-filter based stochastic-multiobjective network programming analysis of inundation evacuation planning is presented in Fig. 1.*

In order to understand the proposed method, it would be helpful to add a Figure where all important elements are present (nodes, arcs. Etc.) with their attributes an letters (i, j, k. etc.), not as a flow chart but as a schematic drawing.

**[Authors]:**

**An example of the transportation network model is presented. Notation is listed as follows.**

$r(i, t, s)$     *Stay in node  i  at time  t  in scenario  s*

$x(i, j, t, s)$     *Transportation from  i  to  j  at time  t  in scenario  s*

$x(i, k, t, s)$     *Transportation from  i  to shelter  k  at time  t  in scenario  s*

[Figure]

Line 94. "...are conducted and displayed on the GIS..." has the software been developed in a GIS? If so, which programming language, which GIS platform? Is it available on web?

**[Authors]:**

**The results are displayed on a free and open-source platform, Quantum GIS (QGIS). The inundation evacuation plans for uncertain scenarios are plotted in Figs. 5-9. Details are added in the manuscript.**

Line 101. What do the authors mean by "...The model also considers uncertain

inundation scenarios"? Flooding scenarios/areas with a very low probability of occurrence. If I understood, the evacuation model does not define the flooding area, that process is conducted by the flood models.

If so, the evacuation model should evacuated the affected areas or whatever zone given as an input data, no matter how uncertain it is. Please, make more clear this question because it is not clear if the authors are referring to the flooding scenario probabilities or the evacuation scenario probabilities.

**[Authors]:**

**First, flooding scenarios are simulated by using the HEC-RAS with uncertain factors. Next, a stochastic-multiobjective evacuation model with two optimization stages is proposed for evacuation investment and routing. In the first stage, decision maker decides expected optimal solution for shelter and transportation capacity expansion under flooding uncertainty. In the second stage, optimal evacuation routing is determined for each inundation scenario. The manuscript is modified as follows.**

> *Incorporating the HEC-RAS model with uncertain factors simulates inundation scenarios, including possible damage to residential areas and transportation networks. The optimization model with two optimization stages is proposed for evacuation investment and routing. In the first stage, authority decides expected optimal solution for shelter and transportation capacity expansion under flooding uncertainty. In the second stage, optimal evacuation routing is determined for each inundation scenario. The notation $s$ and $P(s)$ represent a stochastic scenario and its corresponding probability for uncertainty analysis.*

Line 100. Why are the arcs bidirectional?

**[Authors]:**

**This research assumes roads of transportation network are bidirectional. Then our proposed optimization model determines the optimal evacuation direction for each road. To make it clear, the paragraphs is modified as follows.**

> *In this paper, $i$ and $j$ indicate transportation network nodes, and $k$ represents shelter nodes. Arcs represent transportation roads connecting two nodes; each road has its own transportation capacity. The roads can be traveled both directions so arcs of transportation network are assumed to be bidirectional. The optimal evacuation direction of each arc is determined by the optimization model.*

Line 103. "...decisions for shelter and transportation capacity..." Who make the

decision to expand shelters and roads capacity?, and why?. This is an strategy and it should be linked with a more global emergency plan.

**[Authors]:**

**The stochastic-multiobjective model includes two optimization stages. In the first stage, the authority (local government) decides investment of shelter and transportation capacity. The optimal evacuation plan is determined in the second stage. The paragraph is modified.**

> *The optimization model with two optimization stages is proposed for evacuation investment and routing. In the first stage, authority decides expected optimal solution for shelter and transportation capacity expansion under flooding uncertainty. In the second stage, optimal evacuation routing is determined for each inundation scenario.*

Line 105. "...Following the occurrence of uncertain flooding events" Does it mean that the methodology is only useful for such uncertain flooding events and it cannot be applied for the most common flooding events.

**[Authors]:**

**This research deals with inundation evacuation under uncertainty. However, the framework is still valid for common scenarios without uncertainty. For deterministic flooding event, the optimal evacuation can be solved by inputting one inundation scenario in the model. The paragraph is modified as follows.**

> *The notation $s$ and $P(s)$ represent a stochastic scenario and its corresponding probability for uncertainty analysis. The model can still be applied for deterministic evacuation planning by inputting one deterministic flooding scenario in the model.*

Line 109. "...optimal evacuation plan..." I would say evacuation strategy, and apparently for one threatened area. Are the best routes chosen or just the available routes?

**[Authors]:**

**The sentence is modified as follows.**

> *Eq. (1) determines the optimal facility investment and evacuation routes for each flooding scenario.*

Line 118. "...At the initial time..." What does initial time mean here? Did the authors consider the response time, the warning time, etc.? When the evacuation order was given? Or the evacuation proses starts when everybody is ready. In such a case, the final result of the model should represent the travel time, not the evacuation time.

**[Authors]:**

**The total evacuation time of this research considers authority decision time (DT), notification time (NT), reaction time of the community (RT), and evacuation transportation time (ET). The initial time means the beginning of evacuation transportation time (ET). The sentences are modified.**

Line 120. "...demands that all evacuees must end up in a shelter..." Why must all evacuees end up in a shelter? In real evacuations it is very common that people evacuate with their relatives or in other places, outside the dangerous area, which means they could follow different routes. If all evacuees must reach a shelter, the authors should explain why (see comments above related with the description of the evacuation scenario).

**[Authors]:**

**The model assumes all evacuees must end up in a shelter or some places outside the dangerous area. The manuscript is modified.**

> *Eq. (8) demands that all evacuees must end up in a shelter or some places outside the dangerous area; $k$ is defined as nodes of shelters or other safe places, $k = 1, \dots K$.*

$$\sum_t \sum_{k,i \in nb(k)} x(i,k,t,s) = \sum_i^n IN(i,s) \qquad \forall k, s \qquad (8)$$

3. Results and discussion

Line 168. "...is applied to simulate uncertain inundation scenarios..." Probably I do not understand well, but why are the results of the uncertain inundation scenarios compared to the normal behaviour of the river during a flood? Are they specific or common flooding scenarios? Were the flooding areas assessed using only flood models or they also include real scenarios based on past events?

**[Authors]:**

**The flooding scenarios are simulated by HEC-RAS model under uncertainty of upstream flow, downstream water level, and channel roughness coefficient. The simulated water stages of the Jingmei River are compared with the levee to determine potential inundation locations. The paragraph is modified as follows.**

> *In this study, the HEC-RAS hydraulic model is applied to simulate uncertain inundation scenarios of the Jingmei River. The cross section geometry and characteristics of Jingmei River are obtained from Taiwanese Water Resources Agency. The accuracy of the HEC-RAS model's representation of the real system has been validated. The validation shows that water inflow of 1554 cms, water level of 12.81 m, and manning's roughness coefficient of 0.025 yields accurate water level simulation of*

*Jingmei River.*

> *The HEC-RAS hydraulic model is used to simulate uncertain water stage of the Jingmei River in Muzha. Uncertain upstream flow (0%, ±7%, ±14%), downstream water level (0%, ±6%, ±12%), and channel roughness coefficient (ranging from 0.013 to 0.045 by interval of 0.002) are considered in the model. Fig. 3 plots uncertain simulation of water stage of the Jingmei River. In this study, potential inundation overflow locations are determined by comparing the water stage and levee height.*

Line 173. "...The transportation network system of Muzha is plotted.." Does the evacuation model address the total transportation network or just the main roads? That should be explained in the methodology and highlighted here.

**[Authors]:**

**The proposed evacuation model considers the total transportation roads. The details are explained and the paragraph is modified as follows.**

> *The transportation network system of Muzha is plotted in Fig. 2. The transportation network considers all transportation roads. There are 1079 arcs and 774 nodes for the network. All of the arcs are bidirectional in the traffic network. Shelters accommodate people and save evacuees from danger. Most of the shelters in Muzha are schools or regional activity centers, because these places often incorporate an auditorium and they are convenient locations for delivering to. According to the Disaster Prevention and Protection Organization of Taiwan, there are 17 shelters in the study site with various capacities. The capacities range from 10 to 300 people, and the total of 1277 residents in this area can be accommodated. Shelter locations and capacities are represented by grey circles in Fig. 2.*

Line 175. About shelters, are they known by people at risk? Have they been included in an evacuation plan? Do people know the shelter they should go or is it a free choice? Please, be more specific in the description of the evacuation model you are trying to simulate an how this kind of emergencies are managed in Muzha.

**[Authors]:**

**The proposed stochastic-multiobjective model optimizes evacuation investment and routing. This research assumes that evacuees have the complete information about shelter capacity status and people would follow authority's evacuation plan. The paragraphs are modified as follows.**

> *The evacuation procedures including authority decision, announcement, community reaction, and evacuation. The research*

*develops a framework to analyze evacuation plans for deterministic and uncertain flooding events; then optimal evacuation plan for community is presented. This research assumes that evacuees have the complete information about shelter capacity status and people would follow authority's evacuation plan.*

*The optimization model with two optimization stages is proposed for evacuation investment and routing. In the first stage, authority decides expected optimal solution for shelter and transportation capacity expansion under flooding uncertainty. In the second stage, optimal evacuation routing is determined for each inundation scenario.*

Line 178. "...Shelter locations and capacities..." Is the maximum capacity shown? Is the capacity included in the management of the global emergency plan as a key factor?

**[Authors]:**

**The capacity information of shelters is displayed. The capacity management is included in the global emergency evacuation plan.**

**Table 1. Location and capacity of shelters in Muzha.**

| Shelter | Address | Capacity (people) |
|---|---|---|
| Zhongshun Temple | No. 13, Zhonglun Rd., Muzha | 10 |
| Muzha District Activity Center | No. 3, Ln. 13, Baoyi Rd., Muzha | 37 |
| Muxin District Activity Center | No. 4, Ln. 310, Sec. 3, Muxin Rd., Muzha | 34 |
| Youngjian District Activity Center | No. 177, Sec. 1, Muzha Rd., Muzha | 38 |
| Auditorium, Administration Center | No. 220, Sec. 3, Muzha Rd., Muzha | 60 |
| Zhongshun District Activity Center | No. 22, Sec. 2, Zhongshun St., Muzha | 19 |
| Shihjian District Activity Center | No. 290-1, Sec. 1, Muzha Rd., Muzha | 20 |
| Zhangxin District Activity Center | No. 22, Yishou St., Muzha | 87 |
| Lixing Junior High | No. 7, Ln. 155, Sec. 3, Muxin Rd., Muzha | 30 |
| Muzha Elementary School | No. 191, Sec. 3, Muzha Rd., Muzha | 155 |

| | | |
|---|---|---|
| Muzha Junior High | No. 12, Ln. 102, Sec. 3, Muzha Rd., Muzha | 30 |
| Youngjian Elementary School | No. 2, Shiyuan Rd., Muzha | 30 |
| Mindaw Elementary School | No. 61, Ln. 138, Sec. 2, Muzha Rd., Muzha | 42 |
| Jingmei Girls High School | No. 312, Sec. 3, Muxin Rd., Muzha | 300 |
| Shihjian Elementary School | No. 4, Sec. 1, Zhongshun St., Muzha | 75 |
| Shihjian Junior High | No. 67, Sec. 7, Xinhai Rd., Muzha | 300 |
| Zhangjiao District Activity Center | No. 45, Hengguang St., Muzha | 10 |

Fig. 4. Please add an arrow to show the flow direction of the river.

**[Authors]:**

**Flow direction of the river is added in the figure as follows.**

[Figure]

**Figure 3.** Shelter and transportation network system of Muzha.

Line 184. "...Based on the HEC-RAS simulation of 425 times..." Given the same water level and levee heights, how the HEC-RAS compute the variation in the results?

**[Authors]:**

**Uncertain factors of upstream flow (0%, ±7%, ±14%), downstream water level (0%, ±6%, ±12%), and channel roughness coefficient (ranging from 0.013 to 0.045 by interval of 0.002) are simulated by the HEC-RAS model. The total simulation time is 425 (=5\*5\*17) time. Details of uncertain simulation are explained in the updated**

**manuscript.**

> *The HEC-RAS hydraulic model is used to simulate uncertain water stage of the Jingmei River in Muzha. Uncertain upstream flow (0%, ±7%, ±14%), downstream water level (0%, ±6%, ±12%), and channel roughness coefficient (ranging from 0.013 to 0.045 by interval of 0.002) are considered in the model. Fig. 3 plots uncertain simulation of water stage of the Jingmei River. In this study, potential inundation overflow locations are determined by comparing the water stage and levee height. Then areas within a 200-meter radius around potential overflow sites are regarded as evacuation zones. Accordingly, Fig. 4 displays the three cases of overflow location and inundation evacuation areas including three cases of Xinhai Road Sec 7, Hengkung Bridge, and Daonan Bridge in Muzha. The probability of each inundation scenario depends on the number of simulation for which the potential water stage exceeds the levee height. Based on the HEC-RAS simulation at each location, the probabilities for three inundation areas (Xinhai Road Sec 7, Hengkung Bridge, and Daonan Bridge) are 0.43, 0.15, and 0.42, respectively.*

Line 186. Why are there shelters located in the evacuation area? Is there any problem with the local authorities in the design of the emergency strategies? Are this shelters considered by the evacuation model as sink nodes?

**[Authors]:**

**The manuscript is modified. People are evacuated to shelter located closed to the inundation area. Shelters located in the flooding area is not available for evacuation.**

> *This case study assumes that people living on the first floor needs to be evacuated; the rest of people living on upper floors are not evacuated due to shelter capacity. The proposed stochastic-multiobjective model can further develop complete evacuation plans for all of people in threatened area by assuming whole area as evacuation places.*

Lines 188. Why people living on higher floors do not need to be evacuated? Is it a question related with the capacity of the evacuation model or an evacuation plan design? Please, explain this important question, because people do not use to do what it is expected during an emergency situation (more people travelling could cause traffic jams) and an evacuation simulation never should be done for crisis management considering just part of the people living in a threatened area.

**[Authors]:**

**We agreed that people do not follow what they are expected to do during an emergency situation. Our current case study calculates average number of people living on lower floors and only evacuates people on lower floors. The proposed stochastic-multiobjective model can further develop complete evacuation plans for all of people in threatened area by assuming whole area as evacuation places. The paragraph is modified as follows.**

> *The probability of each inundation scenario depends on the number of simulation for which the potential water stage exceeds the levee height. Based on the HEC-RAS simulation at each location, the probabilities for three inundation areas (Xinhai Road Sec 7, Hengkung Bridge, and Daonan Bridge) are 0.43, 0.15, and 0.42, respectively. People can be evacuated to six shelters located close to the inundation zones. The evacuation area of Xinhai Road Sec 7 area is 0.315 km$^2$. Hengkung Bridge area is 0.069 km$^2$, and Daonan Bridge area is 0.089 km$^2$. Data of people living on each floor are not available. This case study assumes that people living on the first floor needs to be evacuated; the rest of people living on upper floors are not evacuated due to shelter capacity. The proposed stochastic-multiobjective model can further develop complete evacuation plans for all of people in threatened area by assuming whole area as evacuation places.*

Line 193. "...evacuation times of 55, 12, and 24 minutes??

**[Authors]:**

**Thank you. The sentence is corrected.**

Line 194. "… Shihjian Activity Center is the only shelter that is required to be expanded..." that assuming that no one living above the first floor will make the decision to evacuate.

**[Authors]:**

**People living on lower floors and people living on upper floors are calculated as follows. The paragraph is modified accordingly.**

> *People can be evacuated to six shelters located close to the inundation zones. The evacuation area of Xinhai Road Sec 7 area is 0.315 km$^2$. Hengkung Bridge area is 0.069 km$^2$, and Daonan Bridge area is 0.089 km$^2$. Data of people living on each floor are not available. This research assumes the buildings have five floors on average. Then 1/5 of people living on the first floor needs to be evacuated; the rest of people living on upper floors are not evacuated.*
>
> *The results show that the Shihjian Activity Center is the only shelter*

*that is required to be expanded. Xinhai Road Sec 7 flooding scenario dominates the Shihjian Activity Center expansion, because the scenario involves the largest probability of occurrence, the largest inundation area, and the highest number of people to be evacuated. Since the flooding evacuation area of this scenario contains the highest number of residents, the western area of Muzha is the potential critical zone for evacuation. For comparison, the shelter expansion and evacuation planning for the inundation scenarios in Xinhai Road Sec 7, Hengkung Bridge, and Daonan Bridge areas are plotted in Figs. 5-7.*

Line 219. "...further to shelters..." here, do the authors mean that people have to travel far away to find shelters because the closer ones were not expanded and they are full?.

**[Authors]:**

**The sentence means that people need to travel far away to find shelters while the capacity of closer shelter is not sufficient. The sentence is not clear; the paragraph is modified as follows.**

> **Case 2 puts higher weighting to expansion cost, and evacuation time would receive less weighting. Hence, in Case 2, evacuees for Daonan Bridge need to travel far away to find shelters while the capacity of closer shelter is not sufficient.**

4. Conclusions

Line 239. "...Evacuation planning is analyzed for various disasters..." Analyzed or used?

**[Authors]:**

**The sentence is corrected as follows.**

> *This study analyzes stochastic inundation evacuation planning used for flooding events.*

Line 240. "... inundation evacuation under uncertainty..." Again, does it means the framework developed is not valid for more common scenarios whit less uncertainty.

**[Authors]:**

**The framework developed in this research deals with inundation evacuation under uncertainty. However, the framework is still valid for common scenarios with less uncertainty. More specifically, inundation evacuation without uncertainty can be solved by inputting one flooding scenario in the model. The paragraph is modified as follows.**

*This study analyzes stochastic inundation evacuation planning used for flooding events. The KASMNO model was newly established for iterative prediction, measurement, update, and optimization of stochastic inundation simulation and evacuation.*

*The proposed framework can still be applied for deterministic evacuation planning by inputting one deterministic flooding scenario in the model.*

Line 245. "...and evacuation planning have been presented on the GIS platform...". If the presentation on the GIS is oriented to decision makers, all figures should keep the hazard areas (authors should not delete the hazard limits just to show the results of the evacuation model). Decision makers need to understand the global process and where are located the threatened areas (all of them).

**[Authors]:**

**Decision makers indeed need to know the global process and location the threatened areas. We modified Figs. 2-9 to present complete flooding scenarios, shelter expansion, and evacuation routing.**

---

## Author Comment (AC2) · 14 Dec 2017

Journal: NHESS

Title: Kalman-filter based stochastic-multiobjective network optimization and maximal-distance Latin hypercube sampling for uncertain inundation evacuation planning

MS No.: nhess-2017-248

MS Type: Research article

============================================================

**Anonymous Referee #2**

GENERAL COMMENT

This manuscript introduces new concepts to develop the evacuation scenarios considering the uncertainty of inundation evacuation planning. For inundation evacuation under uncertainty, a Kalman-filter based stochastic-multiobjective network optimization model is implemented. Moreover, a maximal-distance Latin hypercube sampling method is used to simulate uncertain scenarios of flooding. A case study in Muzha, Taiwan is then used to apply the new frameworks of evacuation scenarios. The results of implementing these new methods are then discussed.

Although the manuscripts have presented new frameworks to improve the quality of evacuation scenarios under uncertainty, in general, several points need to be improved: (1) The flow of writing is not well presented. Some parts need to be connected with a proper story. The writing seems to be inconsistent considering the flow of the story; (2) The main concept of evacuation used in this study has not been addressed in the text; (3) the contribution/novelty of this concept compared to the existing methodologies has not been discussed clearly. Subsequently, further revisions need to be carried out.

[Authors]:

**Dear referees and editors,**

**Thank you for your useful comments on our manuscript. We have prepared a detailed point-by-point reply to all of the comments as follows. The manuscript has been modified accordingly. Yours,**

**Ming-Che Hu, Yi-Hsuan Shih, Tsang-Jung Chang**

**(1) The whole manuscript is modified and reorganized. The models, flooding scenarios, assumption, limitation are addressed to improve the flow of writing.**

**(2) The evacuation concept is addressed and the manuscript is modified as follows.**

*Evacuation procedures considered here includes authority decision, announcement, community reaction, and evacuation. To simulate flooding scenarios, maximal-distance Latin hypercube method is used to sample*

*three uncertain factors, including upstream inflow (upper boundary condition), downstream water level (lower boundary condition), and friction resistance of channel. Then potential flooding scenarios are simulated using the HEC-RAS hydraulic model and uncertain factors. The stochastic evacuation model can be further simplified to be a deterministic model by inputting deterministic inundation scenario.    Notice that the proposed model assumes that evacuees have the complete information about shelter capacity status and people would follow authority's evacuation plan. Otherwise, additional time needs to be estimated while people's individual behavior exists under incomplete information cases.*

**(3) The contribution and novelty of this research are compared with previous studies and discussed as follows.**

*This study analyzes stochastic inundation evacuation planning used for flooding events. The KASMNO model was newly established for iterative prediction, measurement, update, and optimization of stochastic inundation simulation and evacuation. Maximal-distance Latin hypercube method has been incorporated with HEC-RAS hydraulic modeling to increase sampling and computational efficiency of uncertain flooding simulation. Accordingly, the tradeoff and uncertainty analysis of evacuation planning was conducted. The flooding scenarios, shelter capacity expansion, and evacuation planning have been presented for decision making.*

SPECIFIC COMMENT
ABSTRACT:
line 12: channel friction resistance uncertainty ==> "uncertainty" can be deleted since you already used UNCERTAIN INUNDATION FACTORS
[**Authors**]:
**The redundant words are deleted and the sentences are modified as follows.**

*First, this research proposes a maximal-distance Latin hypercube sampling method to seek maximal space-filling sampling in uncertain flooding factor space. Uncertain inundation factors including upstream inflow, downstream water level, and channel friction resistance are considered. Incorporated with the sampling method, HEC-RAS hydraulic model simulates stochastic flooding scenarios.*

Line 15: THE new measurement?
[**Authors**]:

**The sentences are not clear and they are modified as follows.**

> *Kalman-filter method iteratively predicts the flooding state of the next stage. Then prediction and decision are updated according to latest measurements of flooding state. Accordingly, Kalman-filter based stochastic-multiobjective programming determines optimal shelter capacity expansion in the here-and-now stage and the best evacuation planning for each scenario in the wait-and-see stage.*

The story flow of sentences in the last sentence of abstract (line 19-21) is not well connected.

**[Authors]:**

**We reorganized the structure of Abstract. The end of Abstract is modified as follows.**

> *A case study of stochastic inundation evacuation in Muzha, Taiwan, is conducted. Furthermore, tradeoff between shelter expansion and evacuation time are analyzed. The results show decreasing marginal effect of capacity expansion for evacuation time reduction.*

INTRODUCTION

In my opinion, the problem in the second paragraph (starting from line 33) must be moved in the first paragraph. Therefore, the story will be something like this: general problem, specific evacuation problem, issues in the existing evacuation methodologies developed in the current literature, then addressed why your frameworks are necessary.

**[Authors]:**

**We agree with the comments. The first two paragraphs of Introduction are reorganized. The first paragraph begins with introduction of general problem (nature hazards, inundation events, and evacuation planning). The second paragraph addresses the majority of previous evacuation planning studies. The third paragraph discusses our research framework and compares it with previous studies. The first paragraph of Introduction is rearranged as follows. The details of reorganization of Introduction is presented in the revised manuscript.**

> *The first paragraph:*

> *Natural hazards, such as typhoons, hurricanes, and cyclones, lead to heavy rainfall, severe storms, and then possible inundation events. Inundation might result in serious damage to people, property, and facilities (Parker and Fordham, 1996; Rodrigues et al., 2002; Romanowicz and Beven, 2003). Hence, inundation evacuation planning is an important*

*consideration for preventing the loss of life (Li et al., 2012; Parker and Priest, 2012; Hegger et al., 2014; Zhang and Pan, 2014; Wang etal., 2015; Wood etal., 2016; Azam et al., 2017). To achieve inundation evacuation planning, locations and capacities of protection refuges should first be designed and constructed. Subsequently, decision support systems of emergency evacuation must be planned (Barbarosoglu and Arda, 2004; Bird et al., 2009; Taubenböck et al., 2009; Marrero et al., 2010; Yeo and Cornell, 2009; Bozorgi-Amiri et al., 2013; Pourrahmani et al., 2015; Xu et al., 2016; Hou et al., 2017; Liu et al., 2017; Muhammad et al., 2017).*

Line 48: MANY RELATED STUDIES HAVE DISCUSSED EVACUATION TRANSPORTATION SYSTEM PROBLEMs ==> What is the problem? need to discussed clearly so the contributions of this study can be well presented.

**[Authors]:**

**We add some paragraphs discussing the details of related evacuation transportation issues. We also compare our study with previous studies and address the contribution of our study.**

*Many related studies have discussed important issues of evacuation transportation systems including uncertain scenarios, dynamic logistics, multi-objective tradeoff (Yi and Ozdamar, 2007; Stepanov and Smith, 2009; Abdelghany et al., 2014). Evacuation planning involves various uncertain factors, including unpredictable impacts, stochastic intensity, random locations of hazard, and uncertain responses of evacuees (Li et al., 2012; Yao et al., 2009). Because stochastic evacuation planning for uncertain flooding scenarios is an important consideration (Romanowicz and Beven, 2003; Barbarosoglu and Arda, 2004; Bozorgi-Amiri et al., 2013), stochastic programming models provide powerful tools for dealing with uncertain disaster and evacuation planning (Yao et al., 2009).*

LINE55: WHAT are the two stage of programming models? you then explained in Line 63-64 but the sentences in line55 seem to be unfinished.

**[Authors]:**

**Those sentences are modified as follows.**

*Stochastic multi-stage programming models determine stochastic optimal decisions at each stage by optimizing expected objective value. Two stage programming models are the most basic multi-stage model; the two stage models select current decisions at the first stage and decide uncertain future choices at the second stage (Romanowicz and Beven,*

*2003; Barbarosoglu and Arda, 2004; Bozorgi-Amiri et al., 2013). In addition, Li et al. (2012) constructed bi-level programming models to determine the optimal capacity of shelters and evacuation route systems. Kongsomsaksakul et al. (2005) sought optimal locations and investments for shelters.*

*Our KASMNO model has two optimization stages including evacuation capacity expansion and evacuation routing. In the first stage, authority determines optimal expected solutions for shelter and transportation capacity expansion under flooding uncertainty. In the second stage, optimal evacuation routing is solved for each potential inundation scenario.*

METHODOLOGY

First, what type of evacuation that you present in this manuscript? is this self-evacuation process where the people are decided by themselves where they want to evacuate? or is there any pre-existing road/plan that has been developed before? Is there any announcement from the authorities for the evacuation?

[Authors]:

**The evacuation procedures of our study includes authority decision, announcement, community reaction, and evacuation. This study proposes a Kalman-filtered based stochastic-multiobjective network optimization model to determine the optimal evacuation plan for community. The community follows the proposed evacuation plan while evacuation decision of the authority is made. The details are addressed and added in the manuscript as follows.**

*This study establishes a KASMNO model for analyzing both long-term and short-term inundation evacuation planning. This KASMNO model determines (1) long-term shelter and transportation capacity expansion plans for authorities and (2) short-term evacuation routing for evacuees under flooding scenarios. For short-term evacuation procedures, authority decision, announcement, community reaction, and evacuation are considered. To simulate flooding scenarios, maximal-distance Latin hypercube method is used to sample three uncertain factors, including upstream inflow (upper boundary condition), downstream water level (lower boundary condition), and friction resistance of channel. Then potential flooding scenarios are simulated using the HEC-RAS hydraulic model and uncertain factors. The stochastic evacuation model can be further simplified to be a deterministic model by inputting deterministic inundation scenario. Notice that the proposed model assumes that evacuees have the complete information about shelter capacity status and*

*people would follow authority's evacuation plan. Otherwise, additional time needs to be estimated while people's individual behaviour exists under incomplete information cases. The weighting method is used to analyze the tradeoff between the shelter expansion cost and the evacuation time. The uncertain flooding scenarios, the associate evacuation plans, and tradeoff analysis of multiple objectives are conducted and displayed on the GIS platform. The framework of the Kalman-filter based stochastic-multiobjective network programming analysis of inundation evacuation planning is presented in Fig. 1.*

Second, in principle, the evacuation time is calculated by summing initial reaction time (IRT) and evacuation time (ET). Three components are further considered to calculate the initial reaction time (IRT) including institutional decision time (DT), institutional notification time (NT), and reaction time of the community (RT). Do you consider this concept or everything has been included in the multiobjective optimization introduced in this study? If it has been included, please state it clearly.

**[Authors]:**

**Thanks for the comments. The decision time and notification time are not considered in the proposed model. We modify the model to calculate total evacuation time including authority decision time (DT), notification time (NT), reaction time of the community (RT), and evacuation time (ET). The paragraphs and Eq. (1) are modified as follows.**

*At time $t$, there are $x(i,k,t,s)$ people to transport from node $i$ to its neighbor shelter $k$, so the total evacuation time can be calculated as in Eq. (1). Furthermore, the total evacuation in Eq. (1) considers authority decision time (DT), notification time (NT), reaction time of the community (RT), and evacuation transportation time (ET). Eq. (1) determines the optimal evacuation plan by minimizing the expected total time under uncertainty. The total investment cost is computed in Eq. (2). Furthermore, the weighting method multiplies each objective function by a weighting factor and sums up all weighted objective functions. Eqs. (1)-(2) is combined by the weighting method and the weighted multiobjective functions is presented in Eq. (3).*

$$MIN \ \sum_s \big\{ P(s) \times \big( \sum_t \sum_{i \in nb(k)} \sum_k^m [t \cdot x(i,k,t,s)] + DT + NT + RT \big) \big\} \ (1)$$

$$MIN \ C_s \times \sum_k^m \{ y_1(k) \} + C_t \times \sum_{i,j} \{ y_2(i,j) \} \qquad \qquad (2)$$

$$MIN \ C_r \times \sum_s \big\{ P(s) \times \big( \sum_t \sum_{i \in nb(k)} \sum_k^m [t \cdot x(i,k,t,s)] + DT + NT + RT \big) \big\}$$
$$+ C_s \times \sum_k^m \{ y_1(k) \} + C_t \times \sum_{i,j} \{ y_2(i,j) \} \qquad \qquad (3)$$

RESULTS AND DISCUSSION

In my opinion, the manuscript needs to discuss the comparison between the results from your study and the other existing frameworks that have been validated and used for another study/region. In this section, the results from this study have not been validated with the other methodologies/data.

Line 180: why 200-m is the radius of potential overflow? please clarify

**[Authors]:**

**The results of this study is compared and validated with previous studies. The detailed comparison of models, data, results, and contribution are addressed in the revised methodology, results, and conclusions sections.**

**This research determines the potential flooding scenarios based on the simulated overflow and flooding depth. The average flooding area is calculated and 200 m as the radius of potential overflow area.**

Line 184: why the simulation is only 425 times?

**[Authors]:**

**To simulate uncertainty, the model considers uncertain upstream flow (0%, ±7%, ±14%), downstream water level (0%, ±6%, ±12%), and channel roughness coefficient (ranging from 0.013 to 0.045 by interval of 0.002). The total simulation time is 425 time. This study applies maximal-distance Latin hypercube sampling for simulation; the results show that increase of simulation numbers presents similar results.**

> *The HEC-RAS hydraulic model is used to simulate uncertain water stage of the Jingmei River in Muzha. Uncertain upstream flow (0%, ±7%, ±14%), downstream water level (0%, ±6%, ±12%), and channel roughness coefficient (ranging from 0.013 to 0.045 by interval of 0.002) are considered in the model. Fig. 3 plots uncertain simulation of water stage of the Jingmei River. In this study, potential inundation overflow locations are determined by comparing the water stage and levee height. Then areas within a 200-meter radius around potential overflow sites are regarded as evacuation zones. Accordingly, Fig. 4 displays the three cases of overflow location and inundation evacuation areas including three cases of Xinhai Road Sec 7, Hengkung Bridge, and Daonan Bridge in Muzha. The probability of each inundation scenario depends on the number of simulation for which the potential water stage exceeds the levee height. Based on the HEC-RAS simulation at each location, the probabilities for three inundation areas (Xinhai Road Sec 7, Hengkung Bridge, and Daonan Bridge) are 0.43, 0.15, and 0.42, respectively.*

Lines 185: Why only three probability scenarios presented in this section? are there only three cases? please clarify. Why are these three probs chosen?

**[Authors]:**

**The HEC-RAS hydraulic model is used to simulate uncertain water stage of the Jingmei River in Muzha. This research simulates and then selects three most severe flooding scenarios as case studies. However, the proposed stochastic model is very flexible and more uncertain scenarios can be analyzed using our models and framework.**

*The HEC-RAS hydraulic model is used to simulate uncertain water stage of the Jingmei River in Muzha. Potential inundation overflow locations are determined by comparing the water stage and levee height. Fig. 3 plots simulation of water stage and levee heights of the Jingmei River. In this study, areas within a 200-meter radius around potential overflow sites are regarded as evacuation zones. Accordingly, Fig. 4 displays the three cases of overflow location and inundation evacuation areas including three cases of Xinhai Road Sec 7, Hengkung Bridge, and Daonan Bridge in Muzha.*

[Figure]

**Figure 3.** Simulation of water stage, levee height, and potential overflow of Jingmei River.

[Figure]

**Figure 4.** The overflow location and inundation evacuation areas.

Line 188: How to define the people live on higher floors? do you have all the building data in this areas? please explain it clearly.

**[Authors]:**

**People living on lower floors and people living on upper floors are defined as follows. The paragraph is modified accordingly.**

> ***People can be evacuated to six shelters located close to the inundation zones. The evacuation area of Xinhai Road Sec 7 area is 0.315 km². Hengkung Bridge area is 0.069 km², and Daonan Bridge area is 0.089 km². Data of people living on each floor are not available. This research assumes the buildings have five floors on average. Then 1/5 of people living on the first floor needs to be evacuated; the rest of people living on upper floors are not evacuated.***

Line 198: HOT ZONE changes to CRITICAL zone?

**[Authors]:**

**We agree with the comments. The sentence is modified as follows.**

> ***Since the flooding evacuation area of this scenario contains the highest number of residents, the western area of Muzha is the potential critical zone for evacuation.***

Line 208-209: the sentences are not clear. Suggest to re-write.

**[Authors]:**

**There are some confusing sentences and the paragraph is re-write as follows.**

> *In the stochastic-multiobjective inundation planning, shelter expansions are determined in the first stage before inundations occur. For lower costs of shelter expansion ($C_s$), Case 1 presents more shelter investment. Since Xinhai Road Sec 7 has the larger inundation area with more people must be evacuated, the results show that largest expansion of a shelter is in the west of Muzha (Shihjian Activity Center) for Xinhai Road Sec 7. The number of extra people to be covered by expansions is 259 for Case 1. For higher costs of shelter expansion, Case 2 is more difficult to increase the shelter capacity. Hence, the capacity expansion of Shihjian Activity Center reduces to 116 people. In Case 2, less capacity expansion leads to the efficient location of the new shelter (in the midpoint of Xinhai Road Sec 7, Hengkung Bridge, and Daonan Bridge areas). Thus, all evacuation scenarios can be benefit from this expansion.*

The comments for Figs 8-9 need to elaborate on the concept of calculating the evacuation time.

**[Authors]:**

**The comments for Figs. 8-9 are modified as follows.**

> *Figs. 8-9 display the evacuation planning on the transportation network system for low and high expansion cost cases. Case 1 with the lower expansion cost builds additional shelter with capacity of 259 people.*
>
> *Case 2 with higher expansion cost constructs new shelter with less capacity (116 people). In Case 2, the maximal evacuation time for Daonan Bridge area increases to 23 minutes. This is obvious because Case 2 puts higher weighting to expansion cost, and evacuation time would receive less weighting. Hence, in Case 2, evacuees for Daonan Bridge need to travel far away to find shelters while the capacity of closer shelter is not sufficient.*
>
> *The results indicate that a lower weighting of shelter expansion cost tends to increases shelter capacities, rather than evacuating people to more distant shelters. Conversely, the case with a higher weighting for the expansion cost incurs less shelter expansion. Consequently, an increase in shelter expansion weighting (or cost) reduces the motivation for shelter expansion. Thus, evacuees will be required to travel far away to find shelters, rather than the close shelter being expanded. In addition, comparing flooding areas shows that Xinhai Road Sec 7 area dominates the*

*shelter expansion, because it has highest population density and the most people to evacuate.*

CONCLUSIONS
First and second sentences (line 239-240) is not necessary. Suggest to change or delete it.
**[Authors]:**
**The redundant sentences are deleted. The paragraph is modified in the updated manuscript.**

The limitations of this study also need to be explained clearly in this section. Only future studies are presented but it can be connected with the limitations of this study.
**[Authors]:**
**The limitation of this research is addressed in the Conclusion section. The paragraph is modified as follows.**

> *Uncertain inundation evacuation planning becomes more critical as the frequency and impacts of disaster events increases. The limitation of this research is that real-time and internet of things (IoT) systems disaster observation/data are not considered. Dynamic disaster response and evacuation with IoT systems would be useful for disaster evacuation planning. Further, this study has not analyzed and simulated uncertain social and economic impact of various disasters. Future studies should include economic, social, and spatial-temporal analysis for disaster simulation and evacuation planning. In addition, real-time optimal control and stochastic evacuation with IoT system under sequential disaster events, climate change, hydrological, and geological uncertainty should be further investigated.*